# Bayes-Adaptive Simulation-based Search with Value Function Approximation

**Arthur Guez**[*,1,2]          **Nicolas Heess**[2]          **David Silver**[2]          **Peter Dayan**[1]

[*]aguez@google.com          [1]Gatsby Unit, UCL          [2]Google DeepMind

## Abstract

Bayes-adaptive planning offers a principled solution to the exploration-exploitation trade-off under model uncertainty. It finds the optimal policy in belief space, which explicitly accounts for the expected effect on future rewards of reductions in uncertainty. However, the Bayes-adaptive solution is typically intractable in domains with large or continuous state spaces. We present a tractable method for approximating the Bayes-adaptive solution by combining simulation-based search with a novel value function approximation technique that generalises appropriately over belief space. Our method outperforms prior approaches in both discrete bandit tasks and simple continuous navigation and control tasks.

## 1   Introduction

A fundamental problem in sequential decision making is controlling an agent when the environmental dynamics are only partially known. In such circumstances, probabilistic models of the environment are used to capture the uncertainty of current knowledge given past data; they thus imply how exploring the environment can be expected to lead to new, exploitable, information.

In the context of Bayesian model-based reinforcement learning (RL), Bayes-adaptive (BA) planning [8] solves the resulting exploration-exploitation trade-off by directly optimizing future expected discounted return in the joint space of states and beliefs about the environment (or, equivalently, interaction histories). Performing such optimization even approximately is computationally highly challenging; however, recent work has demonstrated that online planning by sample-based forward-search can be effective [22, 1, 12]. These algorithms estimate the value of future interactions by simulating trajectories while growing a search tree, taking model uncertainty into account. However, one major limitation of Monte Carlo search algorithms in general is that, naïvely applied, they fail to generalize values between related states. In the BA case, a separate value is stored for each distinct path of possible interactions. Thus, the algorithms fail not only to generalize values between related paths, but also to reflect the fact that different histories can correspond to the same belief about the environment. As a result, the number of required simulations grows exponentially with search depth. Worse yet, except in very restricted scenarios, the lack of generalization renders MC search algorithms effectively inapplicable to BAMDPs with continuous state or action spaces.

In this paper, we propose a class of efficient simulation-based algorithms for Bayes-adaptive model-based RL which use function approximation to estimate the value of interaction histories during search. This enables *generalization* between different beliefs, states, and actions during planning, and therefore also works for continuous state spaces. To our knowledge this is the first broadly applicable MC search algorithm for continuous BAMDPs.

Our algorithm builds on the success of a recent tree-based algorithm for discrete BAMDPs (BAMCP, [12]) and exploits value function approximation for generalization across interaction histories, as has been proposed for simulation-based search in MDPs [19]. As a crucial step towards this end we develop a suitable parametric form for the value function estimates that can generalize appropriately

across histories, using the importance sampling weights of posterior samples to compress histories into a finite-dimensional feature vector. As in BAMCP we take advantage of *root sampling* [18, 12] to avoid expensive belief updates at every step of simulation, making the algorithm practical for a broad range of priors over environment dynamics. We also provide an interpretation of root sampling as an auxiliary variable sampling method. This leads to a new proof of its validity in general simulation-based settings, including BAMDPs with continuous state and action spaces, and a large class of algorithms that includes MC and TD upates.

Empirically, we show that our approach requires considerably fewer simulations to find good policies than BAMCP in a (discrete) bandit task and two continuous control tasks with a Gaussian process prior over the dynamics [5, 6]. In the well-known pendulum swing-up task, our algorithm learns how to balance after just a few seconds of interaction. Below, we first briefly review the Bayesian formulation of optimal decision making under model uncertainty (section 2; please see [8] for additional details). We then explain our algorithm (section 3) and present empirical evaluations in section 4. We conclude with a discussion, including related work (sections 5 and 6).

## 2 Background

A Markov Decision Processes (MDP) is described as a tuple $M = \langle S, A, \mathcal{P}, \mathcal{R}, \gamma \rangle$ with $S$ the set of states (which may be infinite), $A$ the *discrete* set of actions, $\mathcal{P} : S \times A \times S \to \mathbb{R}$ the state transition probability kernel, $\mathcal{R} : S \times A \to \mathbb{R}$ the reward function, and $\gamma < 1$ the discount factor. The agent starts with a prior $P(\mathcal{P})$ over the dynamics, and maintains a posterior distribution $b_t(\mathcal{P}) = P(\mathcal{P}|h_t) \propto P(h_t|\mathcal{P})P(\mathcal{P})$, where $h_t$ denotes the history of states, actions, and rewards up to time $t$.

The uncertainty about the dynamics of the model can be transformed into certainty about the current state inside an augmented state space $S^+ = \mathcal{H} \times S$, where $\mathcal{H}$ is the set of possible histories (the current state also being the suffix of the current history). The dynamics and rewards associated with this augmented state space are described by

$$\mathcal{P}^+(h, s, a, has', s') = \int_{\mathcal{P}} \mathcal{P}(s, a, s')P(\mathcal{P}|h)\,\mathrm{d}\mathcal{P}, \quad \mathcal{R}^+(h, s, a) = R(s, a). \quad (1)$$

Together, the 5-tuple $M^+ = \langle S^+, A, \mathcal{P}^+, \mathcal{R}^+, \gamma \rangle$ forms the Bayes-Adaptive MDP (BAMDP) for the MDP problem $M$. Since the dynamics of the BAMDP are known, it can in principle be solved to obtain the optimal value function associated with each action:

$$Q^*(h_t, s_t, a) = \max_{\tilde{\pi}} \mathbb{E}_{\tilde{\pi}} \left[ \sum_{t'=t}^{\infty} \gamma^{t'-t} r_{t'} | a_t = a \right]; \quad \tilde{\pi}^*(h_t, s_t) = \underset{a}{\operatorname{argmax}} \, Q^*(h_t, s_t, a), \quad (2)$$

where $\tilde{\pi} : S^+ \times A \to [0, 1]$ is a policy over the augmented state space, from which the optimal action for each belief-state $\tilde{\pi}^*(h_t, s_t)$ can readily be derived. Optimal actions in the BAMDP are executed greedily in the real MDP $M$, and constitute the best course of action (i.e., integrating exploration and exploitation) for a Bayesian agent with respect to its prior belief over $\mathcal{P}$.

## 3 Bayes-Adaptive simulation-based search

Our simulation-based search algorithm for the Bayes-adaptive setup combines efficient MC search via root-sampling with value function approximation. We first explain its underlying idea, assuming a suitable function approximator exists, and provide a novel proof justifying the use of root sampling that also applies in continuous state-action BAMDPs. Finally, we explain how to model Q-values as a function of interaction histories.

### 3.1 Algorithm

As in other forward-search planning algorithms for Bayesian model-based RL [22, 17, 1, 12], at each step $t$, which is associated with the current history $h_t$ (or belief) and state $s_t$, we plan online to find $\tilde{\pi}^*(h_t, s_t)$ by constructing an action-value function $Q(h, s, a)$. Such methods use simulation to build a search *tree* of belief states, each of whose nodes corresponds to a single (future) history, and estimate optimal values for these nodes. However, existing algorithms only update the nodes that are directly traversed in each simulation. This is inefficient, as it fails to generalize across multiple histories corresponding either to *exactly* the same, or similar, beliefs. Instead, each such history must be traversed and updated separately.

Here, we use a more general simulation-based search that relies on function approximation, rather than a tree, to represent the values for possible simulated histories and states. This approach was originally suggested in the context of planning in large MDPs[19]; we extend it to the case of Bayes-Adaptive planning. The $Q$-value of a particular history, state, and action is represented as $Q(h, s, a; \mathbf{w})$, where $\mathbf{w}$ is a vector of learnable parameters. Fixed-length simulations are run from the current belief-state $h_t, s_t$, and the parameter $\mathbf{w}$ is updated online, during search, based on experience accumulated along these trajectories, using an incremental RL control algorithm (e.g., Monte-Carlo control, Q-learning). If the parametric form and features induce generalization between histories, then each forward simulation can affect the values of histories that are not directly experienced. This can considerably speed up planning, and enables continuous-state problems to be tackled. Note that a search tree would be a special case of the function approximation approach when the representation of states and histories is tabular.

In the context of Bayes-Adaptive planning, simulation-based search works by simulating a future trajectory $h_{t+T} = s_t a_t r_t s_{t+1} \ldots a_{t+T-1} r_{t+T-1} s_{t+T}$ of $T$ transitions (the planning horizon) starting from the current belief-state $h_t, s_t$. Actions are selected by following a fixed policy $\tilde{\pi}$, which is itself a function of the history, $a \sim \tilde{\pi}(h, \cdot)$. State transitions can be sampled according to the BAMDP dynamics, $s_{t'} \sim \mathcal{P}^+(h_{t'-1}, s_{t'-1}, a_{t'}, h_{t'-1}a_{t'}\cdot, \cdot)$. However, this can be computationally expensive since belief updates must be applied at every step of the simulation. As an alternative, we use root sampling [18], which only samples the dynamics $\mathcal{P}^k \sim P(\mathcal{P}\,|h_t)$ once at the root for each simulation $k$ and then samples transitions

**Algorithm 1:** Bayes-Adaptive simulation-based search with root sampling

---

**procedure** Search( $h_t, s_t$ )
  **repeat**
    $\mathcal{P} \sim P(\mathcal{P}\,|h_t)$
    Simulate( $h_t, s_t, \mathcal{P}, 0$ )
  **until** Timeout()
  **return** $\mathrm{argmax}_a\; Q(h_t, s_t, a; \mathbf{w})$
**end procedure**
**procedure** Simulate( $h, s, \mathcal{P}, t$ )
  **if** $t > T$ **then return** $0$
  $a \leftarrow \tilde{\pi}_{\epsilon-\text{greedy}}(Q(h, s, \cdot; \mathbf{w}))$
  $s' \sim \mathcal{P}(s, a, \cdot), r \leftarrow \mathcal{R}(s, a)$
  $R \leftarrow r + \gamma$ Simulate( $has', s', \mathcal{P}, t+1$ )
  $\mathbf{w} \leftarrow \mathbf{w} - \alpha\,(Q(h, s, a; \mathbf{w}) - R)\,\nabla_{\mathbf{w}}Q(h, s, a; \mathbf{w})$
  **return** $R$
**end procedure**

---

according to $s_{t'} \sim \mathcal{P}^k(s_{t'-1}, a_{t'-1}, \cdot)$; we provide justification for this approach in Section 3.2.[1] After the trajectory $h_T$ has been simulated on a step, the $Q$-value is modified by updating $\mathbf{w}$ based on the data in $h_{t+T}$. Any incremental algorithm could be used, including SARSA, Q-learning, or gradient TD [20]; we use a simple scheme to minimize an appropriately weighted squared loss $\mathbb{E}[(Q(h_{t'}, s_{t'}, a_{t'}; \mathbf{w}) - R_{t'})^2]$:

$$|\Delta\,\mathbf{w}\,| = \alpha\;(Q(h_{t'}, s_{t'}, a_{t'}; \mathbf{w}) - R_{t'})\,\nabla_{\mathbf{w}}Q(h_{t'}, s_{t'}, a_{t'}; \mathbf{w}), \qquad (3)$$

where $\alpha$ is the learning rate and $R_{t'}$ denotes the discounted return obtained from history $h_{t'}$.[2] Algorithm 1 provides pseudo-code for this scheme; here we suggest using as the fixed policy for a simulation the $\epsilon-$greedy $\tilde{\pi}_{\epsilon-\text{greedy}}$ based on some given $Q$ value. Other policies could be considered (e.g., the UCT policy for search trees), but are not the main focus of this paper.

## 3.2 Analysis

In order to exploit general results on the convergence of classical RL algorithms for our simulation-based search, it is necessary to show that starting from the current history, root sampling produces the appropriate distribution of rollouts. For the purpose of this section, a simulation-based search algorithm includes Algorithm 1 (with Monte-Carlo backups) but also incremental variants, as discussed above, or BAMCP.

Let $\mathcal{D}_t^{\tilde{\pi}}$ be the *rollout distribution* function of forward-simulations that explicitly updates the belief at each step (i.e., using $\mathcal{P}^+$): $\mathcal{D}_t^{\tilde{\pi}}(h_{t+T})$ is the probability density that history $h_{t+T}$ is generated when running that simulation from $h_t, s_t$, with $T$ the horizon of the simulation, and $\tilde{\pi}$ an arbitrary history policy. Similarly define the quantity $\tilde{\mathcal{D}}_t^{\tilde{\pi}}(h_{t+T})$ as the probability density that history $h_{t+T}$ is generated when running forward-simulations *with root sampling*, as in Algorithm 1. The following lemma shows that these two rollout distributions are the same.

**Lemma 1.** $\mathcal{D}_t^{\tilde{\pi}}(h_{t+T}) = \tilde{\mathcal{D}}_t^{\tilde{\pi}}(h_{t+T})$ *for all policies* $\tilde{\pi} : \mathcal{H} \times A \to [0, 1]$ *and for all* $h_{t+T} \in \mathcal{H}$ *of length* $t + T$.

*Proof.* A similar result has been obtained for discrete state-action spaces as Lemma 1 in [12] using an induction step on the history length. Here we provide a more intuitive interpretation of root sampling as an auxiliary variable sampling scheme which also applies directly to continuous spaces. We show the equivalence by rewriting the distribution of rollouts. The usual way of sampling histories in simulation-based search, with belief updates, is justified by factoring the density as follows:

$$p(h_{t+T}|h_t, \tilde{\pi}) = p(a_t s_{t+1} a_{t+1} s_{t+2} \ldots s_{t+T}|h_t, \tilde{\pi}) \tag{4}$$

$$= p(a_t|h_t, \tilde{\pi})p(s_{t+1}|h_t, \tilde{\pi}, a_t)p(a_{t+1}|h_{t+1}, \tilde{\pi}) \ldots p(s_{t+T}|h_{t+T-1}, a_{t+T}, \tilde{\pi}) \tag{5}$$

$$= \prod_{t \le t' < t+T} \tilde{\pi}(h_{t'}, a_{t'}) \prod_{t < t' \le t+T} p(s_{t'}|h_{t'-1}, \tilde{\pi}, a_{t'-1}) \tag{6}$$

$$= \prod_{t \le t' < t+T} \tilde{\pi}(h_{t'}, a_{t'}) \prod_{t < t' \le t+T} \int_{\mathcal{P}} P(\mathcal{P}|h_{t'-1}) \, \mathcal{P}(s_{t'-1}, a_{t'-1}, s_{t'}) \, \mathrm{d}\mathcal{P}, \tag{7}$$

which makes clear how each simulation step involves a belief update in order to compute (or sample) the integrals. Instead, one may write the history density as the marginalization of the joint over history and the dynamics $\mathcal{P}$, and then notice that an history is generated in a Markovian way *if conditioned on the dynamics*:

$$p(h_{t+T}|h_t, \tilde{\pi}) = \int_{\mathcal{P}} p(h_{t+T}|\mathcal{P}, h_t, \tilde{\pi}) p(\mathcal{P}|h_t, \tilde{\pi}) \, \mathrm{d}\mathcal{P} = \int_{\mathcal{P}} p(h_{t+T}|\mathcal{P}, \tilde{\pi}) p(\mathcal{P}|h_t) \, \mathrm{d}\mathcal{P} \tag{8}$$

$$= \int_{\mathcal{P}} \prod_{t \le t' < t+T} \tilde{\pi}(h_{t'}, a_{t'}) \prod_{t < t' \le t+T} \mathcal{P}(s_{t'-1}, a_{t'-1}, s_{t'}) \, p(\mathcal{P}|h_t) \, \mathrm{d}\mathcal{P}, \tag{9}$$

where eq. (9) makes use of the Markov assumption in the MDP. This makes clear the validity of sampling only from $p(\mathcal{P}|h_t)$, as in root sampling. From these derivations, it is immediately clear that $\mathcal{D}_t^{\tilde{\pi}}(h_{t+T}) = \tilde{\mathcal{D}}_t^{\tilde{\pi}}(h_{t+T})$. $\qquad\square$

The result in Lemma 1 does not depend on the way we update the value $Q$, or on its representation, since the policy is fixed for a given simulation.[3] Furthermore, the result guarantees that simulation-based searches will be identical in distribution with and without root sampling. Thus, we have:

**Corollary 1.** *Define a Bayes-adaptive simulation-based planning algorithm as a procedure that repeatedly samples future trajectories* $h_{t+T} \sim \mathcal{D}_t^{\tilde{\pi}}$ *from the current history* $h_t$ *(simulation phase), and updates the Q value after each simulation based on the experience* $h_{t+T}$ *(special cases are Algorithm 1 and* BAMCP*). Then such a simulation-based algorithm has the same distribution of parameter updates with or without root sampling. This also implies that the two variants share the same fixed-points, since the updates match in distribution.*

For example, for a discrete environment we can choose a tabular representation of the value function in history space. Applying the MC updates in eq. (3) results in a MC control algorithm applied to the sub-BAMDP from the root state. This is exactly the (BA version of the) MC tree search algorithm [12]. The same principle can also be applied to MC control with function approximation with convergence results under appropriate conditions [2]. Finally, more general updates such as gradient Q-learning could be applied with corresponding convergence guarantees [14].

### 3.3 History Features and Parametric Form for the $Q$-value

The quality of a history policy obtained using simulation-based search with a parametric representation $Q(h, s, a; \mathbf{w})$ crucially depends on the features associated with the arguments of $Q$, i.e., the history, state and action. These features should arrange for histories that lead to the same, or similar, beliefs have the same, or similar, representations, to enable appropriate generalization. This is challenging since beliefs can be infinite-dimensional objects with non-compact sufficient statistics that are therefore hard to express or manipulate. Learning good representations from histories is also tough, for instance because of hidden symmetries (e.g., the irrelevance of the order of the experience tuples that lead to a particular belief).

We propose a parametric representation of the belief at a particular planning step based on *sampling*. That is, we draw a set of $M$ independent MDP samples or particles $U = \{\mathcal{P}_1, \mathcal{P}_2, \ldots, \mathcal{P}_M\}$ from the current belief $b_t = P(\mathcal{P}|h_t)$, and associate each with a weight $z_m^U(h)$, such that the vector $z^U(h)$ is a finite-dimensional approximate representation of the belief based on the set $U$. We will also refer to $z^U$ as a function $z^U : \mathcal{H} \to \mathbb{R}^M$ that maps histories to a *feature vector*.

There are various ways one could design the $z^U$ function. It is computationally convenient to compute $z^U(h)$ recursively as importance weights, just as in a sequential importance sampling particle filter [11]; this only assumes we have access to the likelihood of the observations (i.e., state transitions). In other words, the weights are initialized as $z_m^U(h_t) = \frac{1}{M} \;\forall m$ and are then updated recursively using the likelihood of the dynamics model for that particle of observations as $z_m^U(has') \propto z_m^U(h)P(s'|a, s, \mathcal{P}_m) = z_m^U(h)\,\mathcal{P}_m(s, a, s')$.

One advantage of this definition is that it enforces a correspondence between the history and belief representations in the finite-dimensional space, in the sense that $z_U(h') = z_U(h)$ if belief$(h) = $ belief$(h')$. That is, we can work in history space *during planning*, alleviating the need for complete belief updates, but via a finite and well-behaved representation of the actual belief — since different histories corresponding to the same belief are mapped to the same representation.

This feature vector can be combined with any function approximator. In our experiments, we combine it with features of the current state and action, $\phi(s, a)$, in a simple bilinear form:

$$Q(h, s, a; \mathbf{W}) = z_U(h)^T \, \mathbf{W} \, \phi(s, a), \tag{10}$$

where $\mathbf{W}$ is the matrix of learnable parameters adjusted during the search (eq. 3). Here $\phi(s, a)$ is a domain-dependent state-action feature vector as is standard in fully observable settings with function approximation. Special cases include tabular representations or forms of tile coding. We discuss the relation of this parametric form to the true value function in the Supp. material.

In the next section, we investigate empirically in three varied domains the combination of this parametric form, simulation-based search and Monte-Carlo backups, collectively known as BAFA (for *B*ayes *A*daptive planning with *F*unction *A*pproximation).

## 4 Experimental results

The discrete *Bernoulli bandit* domain (section 4.1) demonstrates dramatic efficiency gains due to generalization with convergence to a near Bayes-optimal solution. The *navigation task* (section 4.2) and the *pendulum* (section 4.3) demonstrate the ability of BAFA to handle non-trivial planning horizons for large BAMDPs with continuous states. We provide comparisons to a state of the art BA tree-search algorithm (BAMCP, [12]), choosing a suitable discretization of the state space for the continuous problems. For the pendulum we also compare to two Bayesian, but not Bayes adaptive, approaches.

### 4.1 Bernoulli Bandit

Bandits have simple dynamics, yet they are still challenging for a generic Bayes-Adaptive planner. Importantly, ground truth is sometimes available [10], so we can evaluate how far the approximations are from Bayes-optimality.

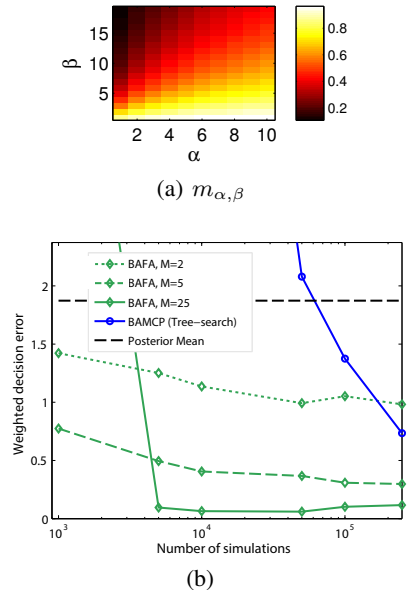

(a) $m_{\alpha,\beta}$

(b)

Figure 1: a) The weights $m_{\alpha,\beta}$ b) Averaged (weighted) decision errors for the different methods as a function of the number of simulations.

We consider a 2-armed Bernoulli bandit problem. We oppose an uncertain arm with prior success probability $p_1 \sim Beta(\alpha, \beta)$ against an arm with known success probability $p_0$. We consider the scenario $\gamma = 0.99, p_0 = 0.2$ for which the optimal decision, and the posterior mean decision frequently differ. Decision errors for different values of $\alpha, \beta$ do not have the same consequence, so we weight each scenario according to the difference between their associated Gittins indices. Define the weight as $m_{\alpha,\beta} = |g_{\alpha,\beta} - p_0|$ where $g_{\alpha,\beta}$ is the Gittins index for $\alpha, \beta$; this is an upper-bound (up to a scaling factor) on the difference between the value of the arms. The weights are shown in Figure 1-a.

We compute the weighted errors over 20 runs for a particular method as $E_{\alpha,\beta} = m_{\alpha,\beta} \cdot P(\text{Wrong decision for } (\alpha, \beta))$, and report the sum of these terms across the range $1 \leq \alpha \leq 10$ and $1 \leq \beta \leq 19$ in Figure 1-b as a function of the number of simulations.

Though this is a discrete problem, these results show that the value function approximation approach, even with a limited number of particles ($M$) for the history features, learns considerably more quickly than BAMCP . This is because BAFA generalizes between similar beliefs.

## 4.2 Height map navigation

We next consider a 2-D navigation problem on an unknown continuous height map. The agent's state is $(x, y, z, \theta)$, it moves on a bounded region of the $(x, y) \in 8 \times 8m$ plane according to (known) noisy dynamics. The agent chooses between 5 different actions, the dynamics for $(x, y)$ are $(x_{t+1}, y_{t+1}) = (x_t, y_t) + l(\cos(\theta_a), \sin(\theta_a)) + \boldsymbol{\epsilon}$, where $\theta_a$ corresponds to the action from this set $\theta_a \in \theta + \{-\frac{\pi}{3}, -\frac{\pi}{6}, 0, \frac{\pi}{6}, \frac{\pi}{3}\}$, $\boldsymbol{\epsilon}$ is small isotropic Gaussian noise ($\sigma = 0.05$), and $l = \frac{1}{3}$m is the step size. Within the bounded region, the reward function is the value of a latent height map $z = f(x, y)$ which is only observed at a single point by the agent. The height map is a draw from a Gaussian process (GP), $f \sim GP(0, \mathcal{K})$, using a multi-scale squared exponential kernel for the covariance matrix and zero mean. In order to test long-horizon planning, we downplay situations where the agents can simply follow the expected gradient locally to reach high reward regions by starting the agent on a small local maximum. To achieve this we simply condition the GP draw on a few pseudo-observations with small negative $z$ around the agent and a small positive $z$ at the starting position, which creates a small bump (on average). The domain is illustrated in Figure 2-a with an example map.

We compare BAMCP against BAFA on this domain, planning over 75 steps with a discount of 0.98. Since BAMCP works with discrete state, we uniformly discretize the height observations. For the state-features in BAFA, we use a regular tile coding of the space; an RBF network leads to similar results. We use a common set of a 100 ground truth maps drawn from the prior for each algorithm/setting, and we average the discounted return over 200 runs (2 runs/map) and report that result in Figure 2-b as a function of the planning horizon ($T$). This result illustrates the ability of BAFA to cope with non-trivial planning horizons in belief space. Despite the discretization, BAMCP is very efficient with short planning horizons, but has trouble optimizing the history policy with long horizons because of the huge tree induced by the discretization of the observations.

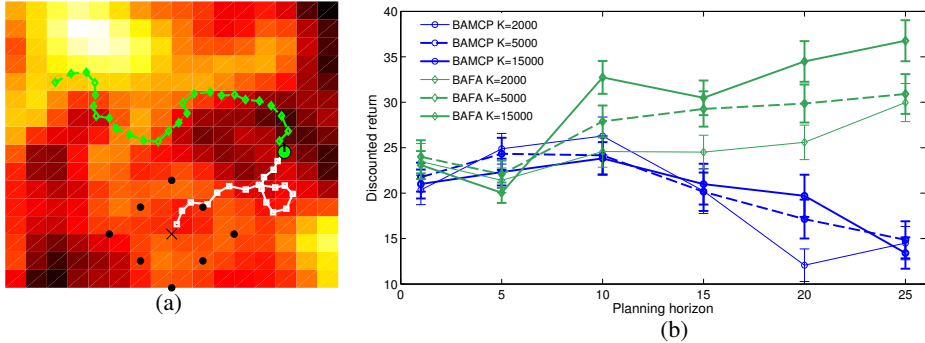

(a)

(b)

Figure 2: (a) Example map showing with the height color-coded from white (negative reward $z$) to black (positive reward $z$). The black dots denote the location of the initial pseudo-observations used to obtain the ground truth map. The white squares show the past trajectory of the agent, starting at the cross and ending at the current position in green. The green trajectory is one particular forward simulation of BAFA from that position. (b) Averaged discounted return (higher is better) in the navigation domain for discretized BAMCP and BAFA as a function of the number of simulations ($K$), and as function of the planning horizon (x-axis).

## 4.3 Under-actuated Pendulum Swing-up

Finally, we consider the classic RL problem in which an agent must swing a pendulum from hanging vertically down to balancing vertically up, but given only limited torque. This requires the agent to build up momentum by swinging, before being able to balance. Note that although a wide variety of methods can successfully learn this task given enough experience, it is a challenging domain for Bayes-adaptive algorithms, which have duly not been tried.

We use conventional parameter settings for the pendulum [5], a mass of 1kg, a length of 1m, a maximum torque of 5Nm, and coefficient of friction of $0.05$ kg m$^2$ / s. The state of the pendulum is $s = (\theta, \dot{\theta})$. Each time-step corresponds to 0.05s, $\gamma = 0.98$, and the reward function is $\mathcal{R}(s) = \cos(\theta)$. In the initial state, the pendulum is pointing down with no velocity, $s_0 = (\pi, 0)$. Three actions are available to the agent, to apply a torque of either $\{-5, 0, 5\}$Nm. The agent does not initially know the dynamics of the pendulum. As in [5], we assume it employs independent Gaussian processes to capture the state change in each dimension for a given action. That is, $s_{t+1}^i - s_t^i \sim GP(m_a^i, \mathcal{K}_a^i)$ for each state dimension $i$ and each action $a$ (where $\mathcal{K}_a^i$ are Squared Exponential kernels). Since there are 2 dimensions and 3 actions, we maintain 6 Gaussian processes, and plan in the joint space of $(\theta, \dot{\theta})$ together with the possible future GP posteriors to decide which action to take at any given step.

We compare four approaches on this problem to understand the contributions of both generalization and Bayes-Adaptive planning to the performance of the agent. BAFA includes both; we also consider two non-Bayes-adaptive variants using the same simulation-based approach with value generalization. In a Thompson Sampling variant (THOMP), we only consider a single posterior sample of the dynamics at each step and greedily solve using simulation-based search. In an exploit-only variant (FA), we run a simulation-based search that optimizes a *state-only* policy over the uncertainty in the dynamics, this is achieved by running BAFA with no history feature.[4] For BAFA, FA, and THOMP, we use the same RBF network for the state-action features, consisting of 900 nodes. In addition, we also consider the BAMCP planner with an uniform discretization of the $\theta, \dot{\theta}$ space that worked best in a coarse initial search; this method performs Bayes-adaptive planning but with no value generalization.

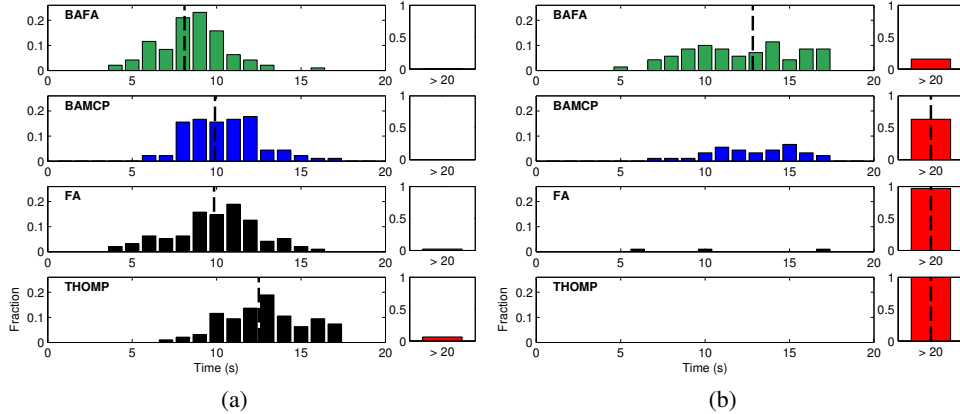

(a)                                                                (b)

Figure 3: Histogram of delay until the agent reaches its first balance state ($|\theta| < \frac{\pi}{4}$ for $\geq$ 3s) for different methods in the pendulum domain. (a) A standard version of the pendulum problem with a cosine cost function. (b) A more difficult version of the problem with uncertain cost for balancing (see text). There is a 20s time limit, so all runs which do not achieve balancing within that time window are reported in the red bar. The histogram is computed with 100 runs with (a) $K = 10000$, or (b) $K = 15000$, simulations for each algorithm, horizon $T = 50$ and (for BAFA) $M = 50$ particles. The black dashed line represents the median of the distribution.

We allow each algorithm a maximum of 20s of interaction with the pendulum, and consider as up-state any configuration of the pendulum for which $|\theta| \leq \frac{\pi}{4}$ and we consider the pendulum balanced if it stays in an up-state for more than 3s. We report in Figure 3-a the time it takes for each method to reach for the first time a balanced state. We observe that Bayes-adaptive planning (BAFA or BAMCP) outperforms more heuristic exploration methods, with most runs balancing before $8.5$s. In the Suppl. material, Figure S1 shows traces of example runs. With the same parametrization of the pendulum, Deisenroth et al. reported balancing the pole after between 15 and 60 seconds of interaction when assuming access to a restart distribution [5]. More recently, Moldovan et al. reported balancing after 12-18s of interaction using a method tailored for locally linear dynamics [15].

However, the pendulum problem also illustrates that BA planning for this particular task is not hugely advantageous compared to more myopic approaches to exploration. We speculate that this

is due to a lack of structure in the problem and test this with a more challenging, albeit artificial, version of the pendulum problem that requires non-myopic planning over longer horizons. In this modified version, balancing the pendulum (i.e., being in the region $|\theta| < \frac{\pi}{4}$) is either rewarding ($\mathcal{R}(s) = 1$) with probability $0.5$, or costly ($\mathcal{R}(s) = -1$) with probability $0.5$; all other states have an associated reward of $0$. This can be modeled formally by introducing another binary latent variable in the model. These latent dynamics are observed with certainty if the pendulum reaches any state where $|\theta| \geq \frac{3\pi}{4}$. The rest of the problem is the same. To approximate correctly the Bayes-optimal solution in this setting, the planning algorithm must optimize the belief-state policy *after* it simulates observing whether balancing is rewarding or not. We run this version of the problem with the same algorithms as above and report the results in Figure 3-b. This hard planning problem highlights more clearly the benefits of Bayes-adaptive planning and value generalization. Our approach manages to balance the pendulum more 80% of the time, compared to about 35% for BAMCP, while THOMP and FA fail to balance for almost all runs. In the Suppl. material, Figure S2 illustrates the influence of the number of particles $M$ on the performance of BAFA.

## 5   Related Work

Simulation-based search with value function approximation has been investigated in large and also continuous MDPs, in combination with TD-learning [19] or Monte-Carlo control [3]. However, this has not been in a Bayes-adaptive setting. By contrast, existing online Bayes-Adaptive algorithms [22, 17, 1, 12, 9] rely on a tree structure to build a map from histories to value. This cannot benefit from generalization in a straightforward manner, leading to the inefficiencies demonstrated above and hindering their application to the continuous case. Continuous Bayes-Adaptive (PO)MDPs have been considered using an online Monte-Carlo algorithm [4]; however this tree-based planning algorithm expands nodes uniformly, and does not admit generalization between beliefs. This severely limits the possible depth of tree search ([4] use a depth of 3).

In the POMDP literature, a key idea to represent beliefs is to sample a finite set of (possibly approximate) *belief points* [21, 16] from the set of possible beliefs in order to obtain a small number of (belief-)states for which to backup values offline or via forward search [13]. In contrast, our sampling approach to belief representation does not restrict the number of (approximate) belief points since our belief features ($z(h)$) can take an infinite number of values, but it instead restricts their *dimension*, thus avoiding infinite-dimensional belief spaces. Wang et al.[23] also use importance sampling to compute the weights of a finite set of particles. However, they use these particles to discretize the model space and thus create an approximate, discrete POMDP. They solve this offline with no (further) generalization between beliefs, and thus no opportunity to re-adjust the belief representation based on past experience. A function approximation scheme in the context of BA planning has been considered by Duff [7], in an offline actor-critic paradigm. However, this was in a discrete setting where counts could be used as features for the belief.

## 6   Discussion

We have introduced a tractable approach to Bayes-adaptive planning in large or continuous state spaces. Our method is quite general, subsuming Monte Carlo tree search methods, while allowing for arbitrary generalizations over interaction histories using value function approximation. Each simulation is no longer an isolated path in an exponentially growing tree, but instead value backups can impact many non-visited beliefs and states. We proposed a particular parametric form for the action-value function based on a Monte-Carlo approximation of the belief. To reduce the computational complexity of each simulation, we adopt a root sampling method which avoids expensive belief updates during a simulation and hence poses very few restrictions on the possible form of the prior over environment dynamics.

Our experiments demonstrated that the BA solution can be effectively approximated, and that the resulting generalization can lead to substantial gains in efficiency in discrete tasks with large trees. We also showed that our approach can be used to solve continuous BA problems with non-trivial planning horizons without discretization, something which had not previously been possible. Using a widely used GP framework to model continuous system dynamics (for the case of a swing-up pendulum task), we achieved state-of the art performance.

Our general framework can be applied with more powerful methods for learning the parameters of the value function approximation, and it can also be adapted to be used with continuous actions. We expect that further gains will be possible, e.g. from the use of bootstrapping in the weight updates, alternative rollout policies, and reusing values and policies between (real) steps.

## Footnotes

[1]For comparison, a version of the algorithm without root sampling is listed in the supplementary material.

[2]The loss is weighted according to the distr. of belief-states visited from the current state by executing $\tilde{\pi}$.

[3] Note that, in Algorithm 1, $Q$ is only updated *after* the simulation is complete.

[4]The approximate value function for FA and THOMP thus takes the form $Q(s, a) = \mathbf{w}^T \phi(s, a)$.

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
