[Supplementary Material · supp.pdf]

# Supplementary Material

Bayes-Adaptive Simulation-based Search with Value Function Approximation

Arthur Guez, Nicolas Heess, David Silver, Peter Dayan

## 1 BAFA without root sampling

---

**Algorithm 2:** Bayes-Adaptive simulation-based search (no root sampling)

---

1: **procedure** Search( $h_t, s_t$ )
2:     **repeat**
3:         Simulate( $h_t, s_t, 0$ )
4:     **until** Timeout()
5:     **return** $\operatorname{argmax}_a Q(h_t, s_t, a; \mathbf{w})$
6: **end procedure**
7: **procedure** Simulate( $h, s, t$ )
8:     **if** $t > T$ **then return** $0$
9:     $a \leftarrow \tilde{\pi}_{\epsilon-\text{greedy}}(Q(h, s, \cdot; \mathbf{w}))$
10:    $s' \sim \mathcal{P}^+(h, s, a, ha\cdot, \cdot)$
11:    $r \leftarrow \mathcal{R}(s, a)$
12:    $R \leftarrow r + \gamma$ Simulate( $has', s', t+1$ )
13:    $\mathbf{w} \leftarrow \mathbf{w} - \alpha \left( Q(h, s, a; \mathbf{w}) - R \right) \nabla_{\mathbf{w}} Q(h, s, a; \mathbf{w})$
14:    **return** $R$
15: **end procedure**

---

Algorithm 2 illustrates the vanilla version of online Sample-Based Planning using Monte-Carlo control without root sampling. Line 10 requires sampling from $\mathcal{P}^+$, a transition in the augmented space which integrates over the dynamics in the posterior distribution. We avoid these expensive operations at every step of the simulation with the root sampling formulation.

## 2 Representing the Value Function

It is known that the value function for the BAMDP is convex as a function of the belief for a particular state [1, 3] (it is piecewise linear if the horizon is finite and the state and action spaces are discrete). Suppose, for simplicity, that states and beliefs are represented exactly (i.e., for example assuming discrete states and $z_U(h) = b(h)$ ), then the bilinear form we introduced in Section 3.3 to represent the value function approximates the true convex value function (for a given state as a function of the belief) with a single linear function: $Q(h, s, a; \{\mathbf{w}_s\}) = \langle b(h), \mathbf{w}_s \rangle$. In general, this is not enough to represent exactly the true value function, but our experiments suggest that it is enough to reason approximately about the consequences of future beliefs.

We have also experimented with an alternative parametric form, an approximately piecewise linear form that combines multiple hyperplanes via a softmax:

$$Q(h, s, a; \{\mathbf{W}_i\}) = \sqrt[k]{\sum_i^I \left( z_U(h)^T \mathbf{W}_i \, \phi(s, a) \right)^k}, \tag{1}$$

inspired by the work of Parr and Russell in the contex of POMDPs [2]. The constants $k$ and $I$ are fixed parameters that trade-off computation and accuracy against the number of learnable parameters (the bilinear form is recovered from the soft-max form using $k = I = 1$). Given sufficient components, this form should be able to represesent the true value function arbitrarily closely. However, in our experiments with this more general form, this advantage was outweighed by its computational complexity, and it performed poorly in practice.

# 3 BAFA implementation details

## 3.1 Learning Rate Schedule

The learning rate schedule we employed for all experiment is $\alpha(n) = a_0 \frac{(n_0+1)}{(n_0+n)}$, where $n$ is the number of weight updates, $a_0$ is the initial learning rate, and $n_0$ influences the speed of decay. We did not try to heavily optimize $n_0$ and $a_0$ for each domain, we only hand-tuned them to avoid divergence or too slow learning - we used the same values for the navigation and the pendulum task (detailed below).

## 3.2 Reusing Particles and Learned Weights

To avoid restarting learning from scratch at every step, we try to reuse the particles $U$ from the previous step and warm-start weight learning from the corresponding learned values in the previous step.

To know whether the set of particles $U_{t-1}$ can still be useful for the current planning step $t$ (i.e., whether the particle set is not degenerate), we compute an estimate of the effective number of particles:

$$N_{\text{eff}} = \frac{1}{\sum_m z_m^{U_{t-1}}(h_t)^2}. \tag{2}$$

If $N_{\text{eff}} < \frac{M}{3}$, then we resample new particles for $U_t$ and reinitialize the weights. Otherwise, we set $U_t = U_{t-1}$ and start learning from the previously learned $\mathbf{W}$.

# 4 Experimental Details

## 4.1 Bandit Domain

In the bandit domain, we set $a_0 = 2.5 \cdot 10^{-3}, n_0 = 2 \cdot 10^5$ for the learning rate schedule. For the exploration parameter for Monte-Carlo control, we set $\epsilon = 2 \cdot 10^{-2}$ to obtain convergence to a near-optimal policy, but different values obtain similar results and mostly affect the distance to optimality after convergence.

Note that there are not state features for this domain, and no discretization is needed for BAMCP since the observations are discrete.

## 4.2 Navigation Domain

The multi-scale kernel $\mathcal{K}$ is a sum of two Squared Exponential kernels $(k_\sigma(x, x') = \exp(\frac{||x-x'||^2}{2\sigma^2}))$ with different length scales: $\mathcal{K}(x, x') = k_{\sigma_1}(x, x') + k_{\sigma_2}(x, x')$, where $\sigma_1 = 0.75$ and $\sigma_2 = 1.5$. In addition, some independent Gaussian observation noise is present, with zero mean and standard deviation $\sigma_n = 0.2$.

Since we cannot store exact samples from a Gaussian Process (it is infinite dimensional), we compute the posterior mean and covariance for the height according to standard formula for a set of 256 points evenly distributed in the 2-D position space. These are then used as an approximation to generate MDP dynamics for this map sample.

The state features $\phi(s)$ is a one-hot vector, obtained by binning the pose space $(x, y, \theta)$ into $D = 1024$ bins ($16 \times 16 \times 4$, uniformly for each dimension). The state-action feature vector $\phi(s, a)$ is then a vector composed of $A+1$ $D-$dimensional subvectors. Each is set to 0 except for the $a$-th subvector and the last subvector, which are both set to $\phi(s)$ (the last, action-independent, subvector is there to allow generalization across actions). As we also noted in the text, an RBF network can also be used here for similar results.

The discretization for BAMCP is made more efficient by only branching on a one-dimensional quantity: the observed reward $z$. This avoids branching on the agent's pose, something which is already approximately captured by the exact encoding of the history in BAMCP. We uniformly

segment heights with $0.5$ increments between $-15$ and $15$; this seemed to be what worked best for the number of simulations we are using in this domain.

We set $a_0 = 5 \cdot 10^{-2}, n_0 = 3 \cdot 10^5$ for the learning rate schedule. We used a more aggressive exploration rate $\epsilon = 2 \cdot 10^{-1}$ since we were more concerned about not exploring enough during search than fine-tuning a near-optimal policy.

Figure S1: Histogram of delay until the agent reaches its first balance state ($|\theta| < \frac{\pi}{4}$ for $\geq$ 3s). The algorithm is BAFA, for different values of the number of particles in the belief representation ($M$), in the modified version of the pendulum problem with hidden costs. All the other parameters are as in Figure 3-b in the main text. We observe that at least around 20 particles are needed to obtain some reasonable performance in this domain. Increasing the number of particles past a certain point provides a diminishing return, since it requires more parameter to learn.

### 4.3 Pendulum Domain

We set the GP kernels to a Squared Exponential kernel $\mathcal{K} = l \cdot k_\sigma$, with $\sigma = 1$. In our experiments, the kernel for the velocity dimension ($\dot{\theta}$) is scaled by a factor of $l = 0.75$ and the one for the angle ($\theta$) is scaled by a factor of $l = 0.25$. Some independent Gaussian observation noise is present, with zero mean and standard deviation $\sigma_n = 0.01$. As in all the other domains, all the compared algorithms shared the same parameters for the prior distribution.

We store the GP samples as in the navigation domain above. We also use the same parameters as above for $a_0$ and $n_0$.

The state features $\phi(s)$ (for BAFA, FA, and THOMP) is obtained from $900$ Radial Basis Functions. The centers of these units are uniformly arranged in the state space. Each unit outputs a similarity measure to a $(\theta, \dot{\theta})$ vector according to:

$$\exp\left(\frac{(\pi - ||\theta - \mu_\theta| - \pi|)^2 + (\dot{\theta} - \mu_{\dot{\theta}})^2}{0.1}\right), \tag{3}$$

where $(\mu_\theta, \mu_{\dot{\theta}})$ are the unit's center coordinates. If the similarity is smaller than some small threshold, we set the corresponding entry to 0 in the feature vector in order to rely on sparse vector

computations. The state-action feature vector $\phi(s, a)$ is then obtained from $\phi(s)$ just like described in the section above.

For BAMCP, we discretized the state space uniformly into 900 bins ($30 \times 30$). That was multiplied by two in the hidden cost version of the pendulum to account for the additional binary state component.

Figure S2: Two runs of BAFA on the pendulum domain, in each run this is the first few seconds of interaction of the agent with the domain. The runs are selected to illustrate a typical good run (a) and a typical slower run (b). Top row shows the absolute value of the pendulum angle $\theta$. Bottom row shows the action selection. Dotted line marks the $\frac{\pi}{4}$ region for up-states.