[Reviews · NeurIPS 2014]

Submitted by Assigned_Reviewer_18

This paper deals with Bayes-adaptive model-based reinforcement learning. The underlying principle of this type of approach is to maintain a Bayesian posterior over dynamics (conditioned on past experienced transitions) and to seek at each time step for the action optimizing the related augmented MDP (on state-history meta-states and related meta-dynamics), which is generally an intractable problem (for exact solving). This contribution relies on two previous ideas, simulation-based search (with "root sampling", which avoids updating the belief over dynamics during planning) and value function approximation (which requires introducing proper features for handling histories), combining them to provide a new approach. In addition to this general approach, called BAFA, the authors provide an alternative and more general proof for the validity of root sampling and provide some experimental results.

Overall, the paper is well written and clear, it proposes a sound approach, based on known ideas but combining them smartly (especially for the history features, which seems to be the newest part/the core contribution). My major comment is that experiences, if convincing, are not detailed enough to be reproducible (for example, some meta-parameters are not provided, nor discussed). There might also be some problems with corollary 1 (is the policy really stationary?).

Detailed (mostly minor) comments:
* Line 106, "belief" can be understood as a synonym of "history" in this sentence, which is misleading
* In the pseudo-code, Simulate(...,d) should be Simulate(...,t), the "+" in the parameters update (Q-value) should be a "-"
* Line 103, a weighted squared loss is mentioned, what are the weights? (do not appear in the next equation, nor in the update rule)
* Line 153, it is said that it is necessary to show that root sampling is valid to exploit general convergence results of classical RL. That's right, but is this sufficient?
* Lines 190-192 (and Corollary 1). I agree with this remark, stating that using a fixed policy implies the same distribution of parameters update, in light of Lemma 1. However, this is not what does the provided algorithm (at least as stated in the pseudo-code): the considered policy is epsilon-greedy respectively to Q_w, but the parameters are updated during planning, so the policy is not stationary. The policy should remain fixed for at least each drawn dynamics (if this is actually the case, the pseudo-code should be more clear).
* For the Bernoulli Bandit, has gamma any importance? How does this compare to standard bandit algorithms?
* Generally, for the experiments, more details could be provided (learning rates for the function estimation, more details about parameterizations, etc.). Ideally, they should be reproducible.
* Fig. 2, K already denotes the kernel of the GP, choose another notation?
* In the height map problem, it is rather strange to consider the reward z as a state component. Why is this done?
* When the dynamics distribution is modeled as a GP, cannot the value function be modeled as a GP too (not necessarily straightforward, due the bilinear representation, but maybe an interesting perspective)?
* Some uppercases are missing in the bib (eg, pomdps in ref 20)
Summary: This paper provides a new algorithm for Bayes-adaptive model-based RL, that basically adds value function approximation (thanks to history based features and a bilinear representation) to an existing simulation-based idea. Overall, this is a good paper. My main concerns are about corollary 1 (which may only require a few clarifications) and about the details of the experiments.

Submitted by Assigned_Reviewer_20

The authors present an approach to Bayesian model-based RL that combines simulation-based search with function approximation. Beliefs are represented and updated by particle filtering. The particle weights define the belief features, which, together with state-action features, define a linear function architecture for value approximation. For any given starting belief-state pair, online RL with function approximation (e.g., Sarsa) computes an approximately optimal online policy. Results on a few benchmark problems demonstrate the approach.

This is a solid paper, with some new interesting ideas, in particular the linear value function architecture over belief importance weights and state-action features. The writing is clear and the results look good. Here are some comments:

As shown by Duff and later by Poupart and others, the Bayes-adaptive value function admits a special form (piecewise linear and convex over the belief space). The paper suggests a different parametrization that is linear over belief features. Clearly the linear architecture is easier to work with and can be directly adapted to work with online RL; but how much one loses by adopting a parametrization that is different to the optimal one? Please discuss this point in the revision, and try to convey intuition about the differences of the two parametrizations.

How many particles have been used in the experiments? How sensitive is the algorithm in the number of particles? (clearly a function of the dimension of the problem; please discuss this in the revision)

The sentence "...forestalling..." should be rewritten.

The authors may consider softening statements like "The first tractable approach", etc.
Summary: An approach to Bayes-adaptive MDPs (= Bayesian RL) that combines ideas from Monte Carlo search and RL with function approximation. A solid paper overall. There are some minor issues that the authors should address in a revised version.

Submitted by Assigned_Reviewer_30

Summary:

This paper suggests to apply function approximation to tree-based Bayes RL algorithms to be able to deal with continuous domains without discretization. A simulation-based algorithm with root sampling is presented and compared to standard Bayes algorithms (that use discretization) in three sample domains (bandit, navigation, pendulum).

Evaluation:

The paper combines various approaches from the literature (function approximation, Bayes-adaptive simulation based search, root sampling) to come up with a Bayes algorithm for continuous domains. While this is done well, there are no new methods presented.

I'm not sure whether there are any theoretical guarantees for the suggested method, as I do not understand the implications of the lemma and the corollary, which are not sufficiently well discussed (cf. below).
On the other hand, the experiments show that the use of function approximation is superior to simple discretization for Bayes algorithms (in the bandit and navigation problem), and also show that the method works well in the pendulum problem. However, the considered examples are rather the usual toy problems, and it is not clear how the algorithm would perform in more challenging settings.

Concerning the write-up, the text gives a lot of intuitive explanations wrt differences to prior approaches, but in my view fails to give sufficiently precise descriptions of the methods themselves. Thus, it was not clear to me which part of the algorithm constitutes root sampling and how this is different from an alternative sampling approach (and what it would look like). Thus, I also could not understand the presented analysis. While I could follow the details in the proof of the lemma, I did not understand what it is supposed to show, nor what this implies on a larger scale.

Some Comments/Questions:

- How does Bayes adaptive planning compare to the ABC RL approach of Dimitrakakis & Tziortziotis (ICML 2013)?

- How does BAFA compare to other bandit algorithms (like UCB, Thompson sampling) in the bandit problem?

- In the description of the navigation experiments, it is mentioned that BAMCP "has trouble optimizing ... because of the huge tree induced by the discretization". To me this sounds as if this would rather imply problems wrt computation time and not wrt performance.
Summary: An ok paper on Bayes RL in continous domains that may have some potential. Conceptually it is just a combination of known results, while the conducted experiments seem to provide a bit of progress but are not completely convincing either.
Author Feedback
Author rebuttal: We thank all reviewers for their insightful reviews and helpful comments. We provide a detailed response below:

To Rev18:

* About corrol. 1 "policy should remain fixed for at least each drawn dynamics [...]"

This understanding is correct, and is reflected in the pseudocode as the parameter updates (which changes the policy) are done *after* the Simulate recursion reaches the end - i.e., after all the policy calls have been made for a particular simulation. We will make this explicit in the text.

* Regarding reproducibility: we will add a detailed description of the domains and parameters in the appendix in the final version, as well as a discussion of the sensitivity to parameters.

* Reward as a state component "Why is this done?"

This is simply to fit the BAMDP formalism. Unknown rewards can be represented as a combination of unknown dynamics and known rewards.

* We will also take the other minor comments into account when preparing the revised version.

To Rev20:

* About (bi)linear architecture vs piecewise linear value form.

We have also experimented with an alternative parametric form, an approximately piecewise linear form that combines multiple hyperplanes via a softmax (which, given enough components, could approx. the true value function arbitrarily closely). However, in our experiments, this advantage was outweighed by its computational complexity, and it performed poorly in practice. In the end we decided not to include these results in the paper. But we will discuss the relation to the optimal value representation in the revision.

* About parameters:

We will include a more explicit discussion of the sensitivity to parameters, like number of particles (very briefly, more particles helped but too many also imply a more difficult learning problem). The number of particles is already provided in Fig 1-b for the bandit problem (parameter M), and in Fig 4 for the pendulum problem.

To Rev30:

* Root sampling:

We are solving the BAMDP using forward simulations. Without root sampling, this requires updating the posterior belief at each step of the simulation (cf Eq 1). The belief update can be very costly, and since many steps are typically required for each simulation, this restricts probabilistic models of the environment to the conjugate case, or other very simple choices. This is the common approach in the literature (cf Asmuth 2011, Wang 2005). Instead, the main insight behind root sampling is that these belief updates are not actually required during simulations, but only after performing a real step.

The use of root sampling is separate from the proposed use of function approximation, but it delivers great computational advantages. It allows us to use complex priors over the dynamics such as Gaussian Processes.

Previous work with root sampling was only concerned with the discrete case, our Lemma+Corrol. justify the use of root sampling for continuous-state spaces (the focus of this paper). We also believe this generalized proof is more intuitive than the one provided by Guez 2012.

We will clarify this in the revised version. Also we will list (in appendix) for comparison a 'vanilla' algorithm, without root sampling.

* Comparison to "ABC RL approach of [...]"

Their paper focuses on a very different question, the problem of posterior computation in the special case where the likelihood, P(s,a,s'), is intractable. Our focus is on planning in a Bayesian RL scenario, which is only a secondary consideration in their paper: they recognize that finding the Bayes-optimal solution is intractable and so use a heuristic form of planning: "[...] in this paper we focus on an approximate version of Thompson sampling for reasons of simplicity." In the sequential case, Thompson Sampling is a useful heuristic but has no guarantees with respect to the Bayes-optimal solution, the theoretical guarantees are only in terms of regret (For more details, the following paper discusses some strongly related examples where Thompson Sampling performs poorly: http://arxiv.org/pdf/1402.1958v1)

* "How does BAFA compare to other bandit alg [...]"

For the bandit problem, we showed that BAFA converges to a near-Bayes-optimal solution (i.e., Gittins indices). UCB and Thompson Sampling perform badly when there is a discount factor < 1 (typically over-exploring since they do not take into account the effect of discounting -- which is correctly accounted for by the BA policy).

* "imply problems wrt computation time and not wrt performance."

Bayes-adaptive planning is primarily a computational problem. The solution is (in a Bayesian sense) the best possible performance that can be achieved by any exploration strategy. It can be found by solving a Bellman eq. but unfortunately it is intractable even for a few states (see e.g. Duff 2002). Therefore we care about approximations that scale well with computation.

* "considered examples are rather the usual toy problems"

Even toy problems lead to enormously complex problems in the Bayesian exploration setting that we consider. Please note that these problems are _much_ more complex than prior work on BA planning research: small discrete MDPs, with the most common domain in recent publications being a 5-state chain MDP (cf Munos 2013, Kolter 2009, Poupart 2006, Strens 2000). To our knowledge, approximating the Bayes-optimal exploration strategy in standard domains such as the pendulum has not been considered before due to the computational barrier.

Our approach extends root sampling to the continuous case and more general simulation-based search scenarios, and it proposes a novel parametric function approximator that allows generalization in history space, critical for BAMDPs. Our experiments demonstrate that this has the potential of opening up previously neglected continuous domains to BA planning, and that the same method can be used to realize generalization in large discrete problems.